# Coupling between fast and slow oscillator circuits in *Cancer borealis* is temperature-compensated

**Daniel Powell[†‡], Sara A Haddad[†§], Srinivas Gorur-Shandilya[†], Eve Marder***

Biology Department and Volen Center, Brandeis University, Waltham, United States

**Abstract** Coupled oscillatory circuits are ubiquitous in nervous systems. Given that most biological processes are temperature-sensitive, it is remarkable that the neuronal circuits of poikilothermic animals can maintain coupling across a wide range of temperatures. Within the stomatogastric ganglion (STG) of the crab, *Cancer borealis*, the fast pyloric rhythm (~1 Hz) and the slow gastric mill rhythm (~0.1 Hz) are precisely coordinated at ~11°C such that there is an integer number of pyloric cycles per gastric mill cycle (integer coupling). Upon increasing temperature from 7°C to 23°C, both oscillators showed similar temperature-dependent increases in cycle frequency, and integer coupling between the circuits was conserved. Thus, although both rhythms show temperature-dependent changes in rhythm frequency, the processes that couple these circuits maintain their coordination over a wide range of temperatures. Such robustness to temperature changes could be part of a toolbox of processes that enables neural circuits to maintain function despite global perturbations.

**\*For correspondence:**
marder@brandeis.edu

[†]These authors contributed equally to this work

**Present address:** [‡]Bowdoin College, Brunswick, United States; [§]University of Zurich, Irchel, Zurich, Switzerland

**Competing interests:** The authors declare that no competing interests exist.

## Introduction

Coupled oscillators are common in nervous systems. Although oscillatory circuits may have distinct frequencies and duty cycles, coordination between them is often necessary for proper function (*Nadim et al., 1998*; *Bartos et al., 1999*; *Colgin, 2011*; *Gordon, 2011*; *Jacobson et al., 2013*; *Rojas-Líbano et al., 2014*; *Harris and Gordon, 2015*; *Karalis et al., 2016*; *Tamura et al., 2017*). For example, theta frequency circuits coupled to those in the gamma range are thought to drive both sensory and behavioral processing (*Lisman and Buzsáki, 2008*; *Fujisawa and Buzsáki, 2011*; *Gordon, 2011*; *Lisman and Jensen, 2013*). Memory formation may rely on the phase coupling of discrete populations of neurons with distinct frequency bands in the hippocampus (*Tass et al., 1998*; *Lisman, 2005*; *Montgomery and Buzsáki, 2007*; *Scheffer-Teixeira and Tort, 2016*; *Zheng et al., 2016*). Network uncoupling or improper coupling may lead to disease states and circuit malfunction (*Moran and Hong, 2011*; *Kirihara et al., 2012*; *de Hemptinne et al., 2013*; *Bahramisharif et al., 2016*; *Salimpour and Anderson, 2019*). The coordination of oscillatory circuits, often with distinct temporal features, is therefore key to circuit function and information processing. Because fluctuations in a circuit's environment can impact function (*Fonseca and Correia, 2007*; *Tang et al., 2010*; *Haddad and Marder, 2018*; *Haley et al., 2018*; *Kushinsky et al., 2019*; *He et al., 2020*), it is important to know how robust this coordination is to externally or internally generated perturbations.

The stomatogastric ganglion (STG) of the Jonah crab, *Cancer borealis*, contains two well-characterized, coupled, oscillatory circuits with known connectivity that generate distinct rhythms with different cycle periods (*Coleman et al., 1995*; *Nadim et al., 1998*; *Marder and Bucher, 2007*). The pyloric rhythm (~1 Hz) is driven by a three-neuron bursting pacemaker kernel composed of the single Anterior Burster (AB) neuron and two Pyloric Dilator (PD) neurons (*Nadim et al., 1998*; *Marder and Bucher, 2007*). The gastric mill rhythm (~0.1 Hz) is produced by activating a half-center oscillator

(reciprocal inhibition that produces alternating bursts of activity) between the Lateral Gastric (LG) motor neuron and Interneuron 1 (Int1), and depends on a complex set of mechanisms including electrical junctions and modulatory input (*Coleman and Nusbaum, 1994*; *Coleman et al., 1995*; *Bartos and Nusbaum, 1997*; *Nadim et al., 1998*; *Bartos et al., 1999*; *Marder and Bucher, 2007*; *Stein et al., 2007*). Although the pyloric and gastric mill oscillators independently produce distinct rhythms (*Bartos et al., 1999*), the core generator neurons (AB and LG) are integer-coupled to each other and influence one another (*Coleman and Nusbaum, 1994*; *Marder et al., 1998*; *Nadim et al., 1998*).

In the New England Atlantic Ocean, where *C. borealis* is found, intertidal temperatures normally range from 8°C to 24°C during the summer and 3°C to 18°C in the winter (*Stehlik et al., 1991*). Because temperature affects most biological processes differently, it poses an interesting challenge for the nervous systems of poikilothermic animals that experience substantial temperature changes (*Hazel and Prosser, 1974*; *Zecevic and Levitan, 1980*; *Prosser and Nelson, 1981*; *Burrows, 1989*; *Foster and Robertson, 1992*; *Xu and Robertson, 1994*; *Xu and Robertson, 1996*; *Franz and Ronacher, 2002*; *Garrity et al., 2010*; *Marder et al., 2015*). For example, the timescales of activation and inactivation of ion channels can change ~2 to 20 fold across 10°C ($Q_{10}$s; *Zečević et al., 1985*; *Klöckner et al., 1990*; *Moran et al., 2004*; *Cao and Oertel, 2005*; *Fohlmeister et al., 2010*; *Tang et al., 2010*; *Kang et al., 2012*; *Robertson and Money, 2012*; *Yang and Zheng, 2014*). Because neuron burst generation depends on many interdependent processes, if even a few key processes have $Q_{10}$s that are appreciably different, this could dramatically disrupt burst generation and/or alter the $Q_{10}$ of a neuron's cycle period. Although the phase relationships between different pyloric neurons are stable between 7°C and 23°C (*Tang et al., 2010*; *Haddad and Marder, 2018*), little is known about temperature compensation in the gastric mill oscillator (*Städele et al., 2015*) or how temperature affects coupling between the pyloric and gastric mill rhythms.

A key feature of temperature compensation in the pyloric circuit is that the pacemaker kernel of the pyloric rhythm maintains constant duty cycle across a range of temperatures (*Tang et al., 2010*; *Rinberg et al., 2013*), and this facilitates temperature compensation of neuron phase relationships in the rest of the circuit. Because the pyloric and gastric mill rhythms arise from different processes, it was not clear that the gastric mill rhythm would be phase-compensated as a function of temperature, and whether the relationship between the two rhythms, such as integer coupling, would be maintained as temperature varies.

## Results

### Two rhythmic oscillators driven by different processes

There are two major rhythms produced by neurons in the STG. The pyloric rhythm is constitutively active, with a characteristic period of about 1 s. The gastric mill rhythm can be evoked by stimulating projection neurons found within the commissural ganglia (CoGs) and has a characteristic period of 6 to 10 s (*Beenhakker et al., 2004*). The modulatory commissural neuron 1 (MCN1) and commissural projection neuron 2 (CPN2; *Figure 1a*) evoke a gastric mill rhythm when stimulated or when they are spontaneously active (Materials and methods; *Beenhakker and Nusbaum, 2004*).

The activity of both circuits can be monitored by extracellular recordings from the motor nerves exiting the STG. The gastric mill rhythm can be seen in the lateral gastric nerve (*lgn*), dorsal gastric nerve (*dgn*) and medial ventricular nerve (*mvn*) recordings in *Figure 1b*. These recordings show alternating bursts of activity in the LG and Dorsal Gastric (DG) neurons, with the Gastric Mill (GM) neurons firing at the end of the LG burst. The VD (Ventricular Dilator) and IC (Inferior Cardiac) neurons, seen on the *mvn,* show an envelope of activity alternating with the LG neuron, but also fire in time with the faster pyloric rhythm seen on the lateral ventricular nerve (*lvn*) and pyloric dilator nerve (*pdn; Figure 1b*). Because the AB and PD neurons are strongly electrically coupled, the cycle period of the pyloric pacemaker can be monitored using the *pdn* (*Figure 1b*), which contains only PD neuron action potentials. Similarly, the LG neuron indicates gastric mill rhythm cycle period (*Figure 1b*).

*Figure 1c* is a connectivity diagram of neurons in the STG and shows how MCN1 and CPN2 activate the gastric mill neurons. While CPN2 drives the gastric mill circuit via electrical synapses onto key motor neurons (LG and GM), MCN1 excites all neurons in the STG, including the pyloric pacemaker and gastric mill half-center neurons (LG and Int1; *Figure 1d*). Both pattern-generating circuits

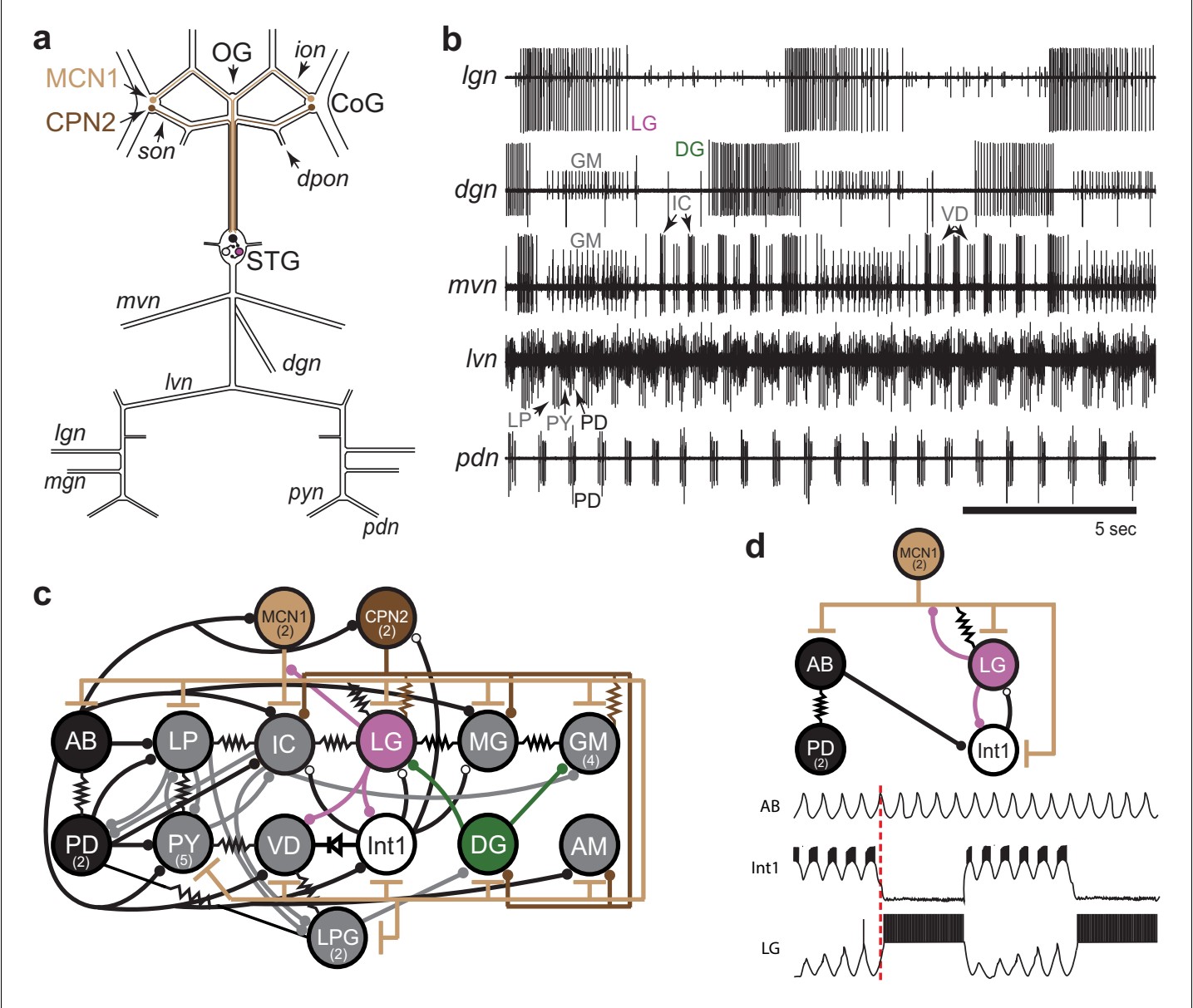

**Figure 1.** STG rhythmic motor patterns. (**a**) Schematic of the STNS showing the locations of salient ganglia, nerves and projection neurons. Individual STG neurons are unambiguously identified on the basis of the motor nerves by which they project to specific stomach muscles. (**b**) Extracellular nerve recordings of both the gastric mill (*lgn*, *dgn*, *mvn*) and pyloric (*lvn*, *pdn*) rhythms. The gastric mill rhythm is seen as the alternating bursts of activity in the LG and DG neurons. The GM neurons are active late in the LG neuron burst. The IC and VD neurons are active in alternation with LG, but also show activity in time with the pyloric rhythm (*lvn*), with the PD neurons shown on the *pdn*. (**c**) Wiring diagram of the gastric mill, pyloric and gastropyloric neurons and the input they receive from MCN1 (tan) and CPN2 (brown; *Marder and Bucher, 2007*; *Blitz, 2017*). LG (pink) and DG (green) are both gastric mill neurons and AB-PD (black) are pyloric neurons. Colors used for these neurons are consistent in all figures. The resistor symbols denote electrical synapses, diode symbol denotes a rectifying junction, chemical inhibitory synapses are shown as filled circles and the T bars indicate chemical excitatory/modulatory synapses from MCN1. Neuron copy number is in parenthesis. (**d**) (Top) Simplified wiring diagram of MCN1 excitation of both pattern generating circuits and AB inhibition of Int1 (white; *Coleman and Nusbaum, 1994*; *Bartos and Nusbaum, 1997*). (Bottom) Cartoon voltage traces of AB, Int1 and LG, as they would appear during a gastric mill rhythm showing the relative burst timing between these neurons. As a consequence of a slow excitation from MCN1 to LG, the envelope of peak disinhibition of LG (from Int1) increases slowly over several cycles, eventually resulting in LG crossing threshold and bursting in time with the AB neuron. Note the slight delays between the peak of AB depolarization, the trough of Int1 inhibition from AB and the subsequent burst of LG as marked by the dashed line.

provide feedback to these projection neurons (*Norris et al., 1994*; *Blitz and Nusbaum, 2008*; *Blitz and Nusbaum, 2012*; *Blitz, 2017*). The interactions between CPN2, MCN1, and LG shape the timing of the gastric mill rhythm along with rhythmic inhibition of Int1 by AB that allows LG to be rhythmically released from inhibition and burst (*Coleman and Nusbaum, 1994*; *Coleman et al., 1995*). The cartoon voltage traces (*Figure 1d*) demonstrate integer coupling between the pyloric and gastric mill circuits, where rhythmic AB inhibition of Int1 results in LG disinhibition and firing following the peak of Int1 inhibition (*Nadim et al., 1998*; *Bartos et al., 1999*).

## Spontaneous gastric mill rhythms span a physiological range of temperatures

While the circuit controlling the pyloric rhythm is temperature-robust from 7°C to 25°C (*Tang et al., 2010*; *Tang et al., 2012*; *Rinberg et al., 2013*; *Soofi et al., 2014*; *Haddad and Marder, 2018*), it remains unclear if the gastric mill circuit could also operate across this same range (*Städele et al., 2015*). We inspected a dataset of extracellular recordings of *dgn* and *lgn* during temperature ramps (*Haddad and Marder, 2018*) and found spontaneous gastric mill activity at temperatures between 11°C and 30°C.

*Figure 2a* shows bursts of LG (*lgn*) and DG (*dgn*) neurons from a single preparation showing spontaneous gastric mill activity at these temperatures. While these recordings show relatively regular gastric mill rhythms at lower temperatures, they were less regular at higher temperatures. *Figure 2b* shows example raster plots of the LG neuron for two preparations with spontaneous gastric mill rhythms. Color indicates the temperature at which that rhythm was observed and can be seen in the raster plots from two preparations across a range of temperature. Preparation 1 intermittently displayed characteristic gastric mill rhythms at both high and low temperatures, but was much more irregular in the middle of the temperature range, whereas preparation 2 was considerably less robust at high temperatures.

Although *Figure 2* only shows two examples of spontaneous gastric mill rhythm, there were a total of nine preparations in which spontaneous rhythms were observed at a variety of different temperatures (*Figure 2—figure supplement 1*). Together, these data indicate that activating at least some combination of endogenous inputs to the gastric mill circuit can generate rhythmic activity across a wide temperature range. Nonetheless, because the cycle period of the gastric mill was often irregular in these spontaneous rhythms, we were not able to assess whether the pyloric and gastric mill circuits remained coupled. Consequently, we chose to reliably activate gastric rhythms by stimulating a known sensory pathway (Materials and methods), to get a more reproducible gastric mill rhythm, of known mechanism, across temperature and across preparations.

## Gastric mill rhythms speed up with increased temperature

We were able to evoke gastric rhythms across the same temperature range at which the pyloric circuit is robust (*Tang et al., 2010*). *Figure 3* shows example recordings of evoked gastric rhythms at 7°C (*Figure 3a*) and 23°C (*Figure 3b*). Raster plots from this preparation for the entire temperature range are shown in *Figure 3c*. Similar to the pyloric circuit, the gastric mill rhythm cycle period decreased with increased temperature from 7°C to 23°C. Although we observed that some of the neuron phase relationships (relative timing of the neurons in each cycle) were altered at 23°C (e.g. between DG and LG in *Figure 3b*), at high temperatures, the gastric mill rhythm generator still produced regular rhythmic activity following stimulation. We were able to evoke gastric rhythms across a temperature range of 7 to 21°C in 9 of 10 preparations.

In some preparations (4 of 10), spontaneous gastric mill rhythms were observed prior to evoking a rhythm. *Figure 3—figure supplement 1a* shows example recordings of both a spontaneous (black) and evoked (blue) gastric mill rhythm from the same preparation (11°C). Three of the four preparations that had spontaneous and evoked rhythms shared similarities in LG spike and burst timing (*Figure 3—figure supplement 1b*) and rhythm cycle period (mean ± SD; *Figure 3—figure supplement 1c*). There are 10 known processes of gastric mill rhythm generation and all but two produce distinct motor patterns from one another (*Heinzel et al., 1993*; *Coleman and Nusbaum, 1994*; *Norris et al., 1994*; *Norris et al., 1996*; *Weimann et al., 1997*; *Blitz et al., 1999*; *Christie et al., 2004*; *Saideman et al., 2007a*; *Saideman et al., 2007b*; *White and Nusbaum, 2011*; *Dickinson et al., 2015*; *Blitz et al., 2019*). The spontaneous rhythms shared attributes with a rhythm

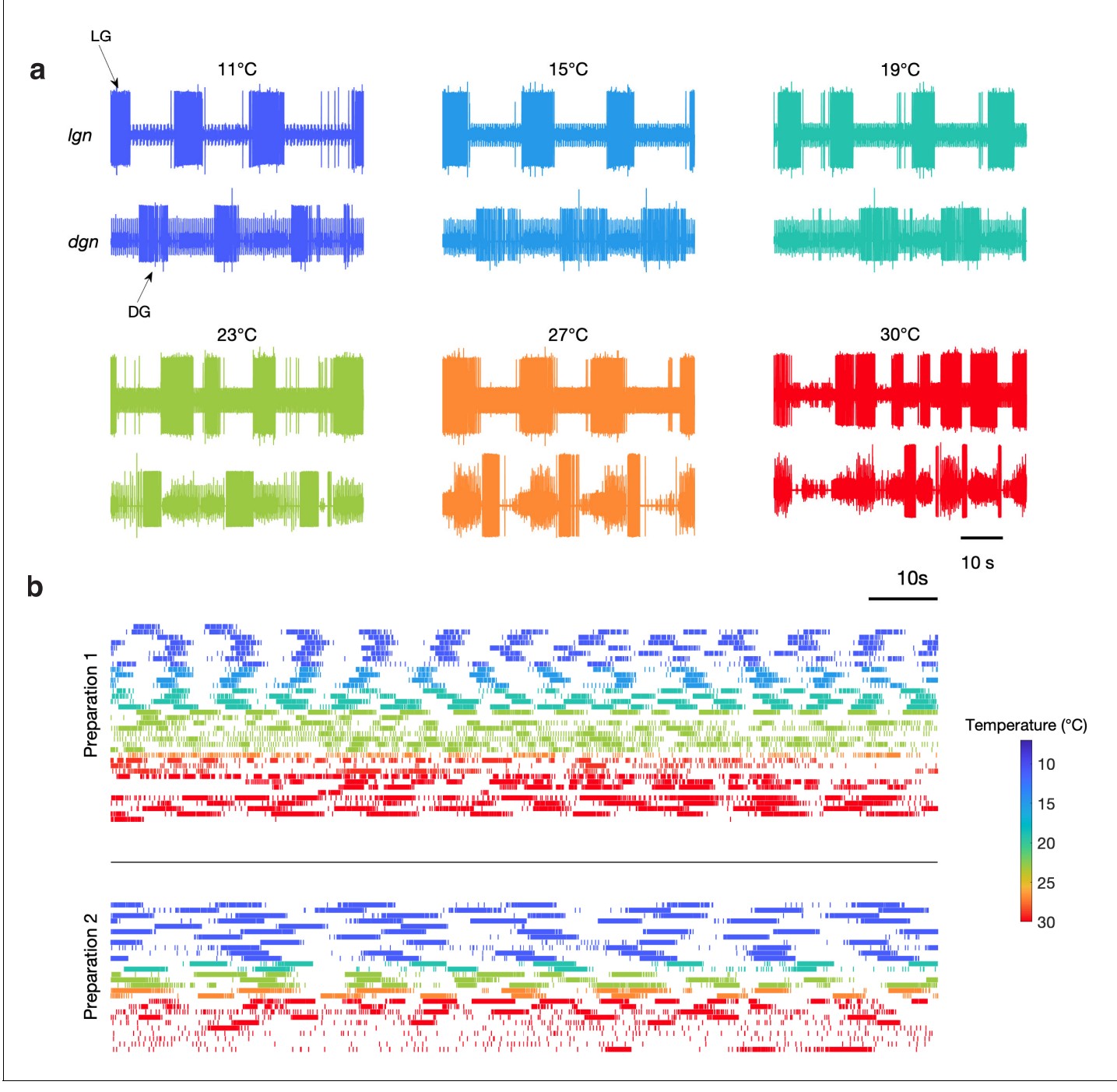

**Figure 2.** Spontaneous gastric mill rhythms over physiological temperature range. (a) Extracellular recordings from the *lgn* and *dgn* at the temperatures indicated using the color code. (b) Spike rasters of LG from two preparations at the indicated colors to show the activity of these preparations over extended periods of time at the indicated temperatures. Traces in (a) and rasters in preparation 2 in (b) are from the same animal. Rasters with the same color are continuous recordings at the same temperature, and time wraps around the end of each raster and continues on the next line. The online version of this article includes the following figure supplement(s) for figure 2:

**Figure supplement 1.** Spontaneous gastric activity in seven additional preparations.

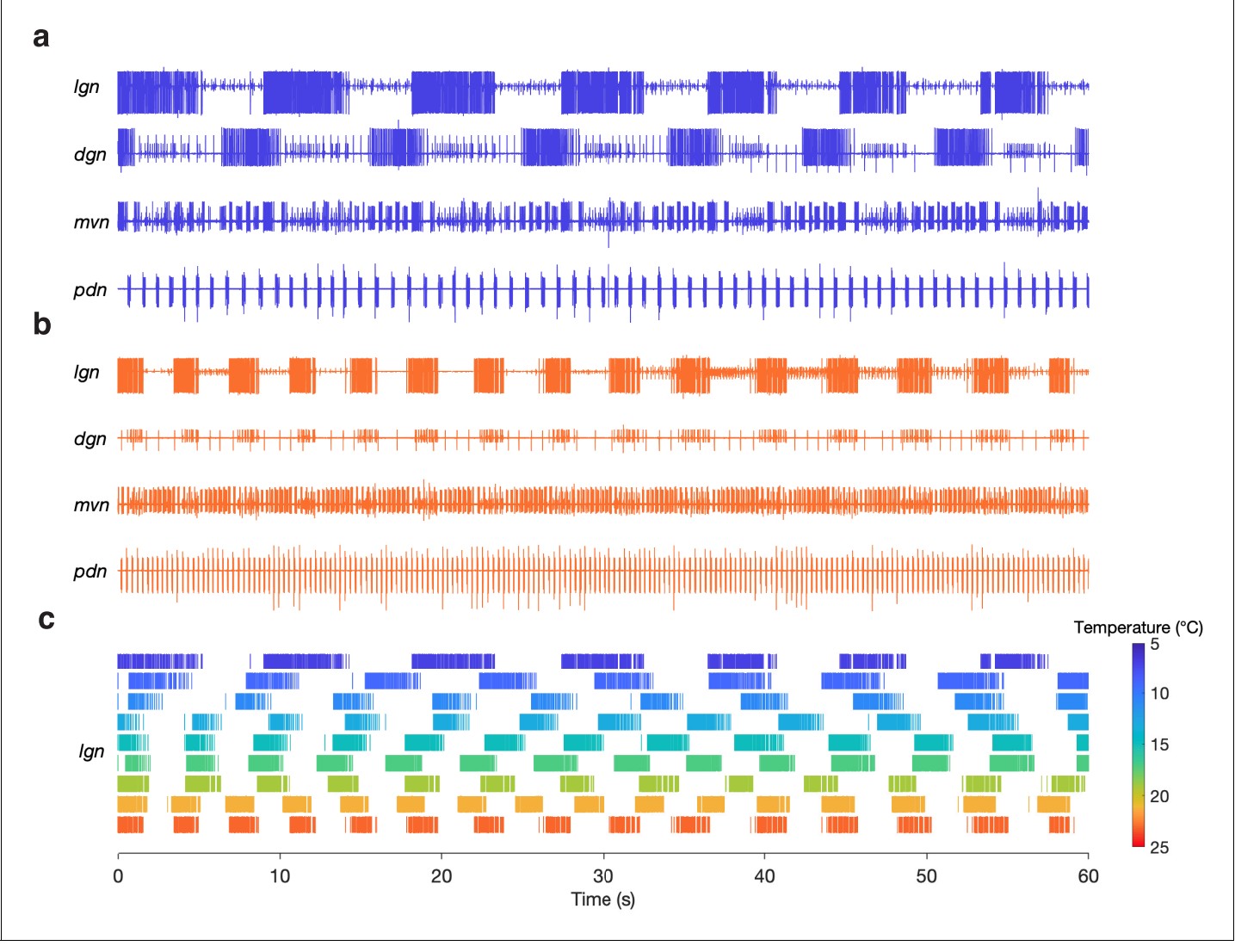

**Figure 3.** Gastric mill rhythms can be evoked by stimulation over physiological temperature range. (**a**) Extracellular recordings from a preparation at 7°C (blue) and (**b**) 23°C (orange). Neuron action potentials described by their relative amplitudes on each nerve: *lgn*: LG (large bursting unit); *dgn*: DG (large, bursting unit; out of phase with LG: only on 7°C trace, has ceased bursting at 23°C), AGR (tonic), GM (small bursting unit; in-phase with LG); *mvn*: IC (large bursting unit), VD (medium bursting unit, antiphase with LG, alternates with IC), GM (small bursting unit; in-phase with LG); *pdn*: PD. (**c**) Spike rasters showing the activity of the LG neuron over the entire temperature range. The first row in (**c**) corresponds to (**a**) and the last row corresponds to (**b**).

The online version of this article includes the following figure supplement(s) for figure 3:

**Figure supplement 1.** Comparison of spontaneous and evoked gastric mill rhythms within the same preparation.

produced by sensory stimulation (Materials and methods). We were curious if evoking this version of gastric mill rhythm would also tolerate temperature change.

## Gastric mill and pyloric cycle periods are similarly temperature-sensitive

The pyloric cycle period decreases ~2-fold with a 10°C increase in temperature (*Tang et al., 2010*). If the pyloric and gastric mill circuits are to be coordinated across temperature, we hypothesized that the gastric mill period might also show a similar temperature dependence. We measured the burst periods of PD neurons (*Figure 4*, black) and LG neurons (*Figure 4*, magenta) in all 10 preparations in which the gastric mill rhythm was evoked by stimulation over a temperature range of 7 to 21°C. *Figure 4a* shows that, in a single preparation, both LG and PD burst periods decrease with increasing temperature, despite significant variability in the LG burst period, due to a slowing down

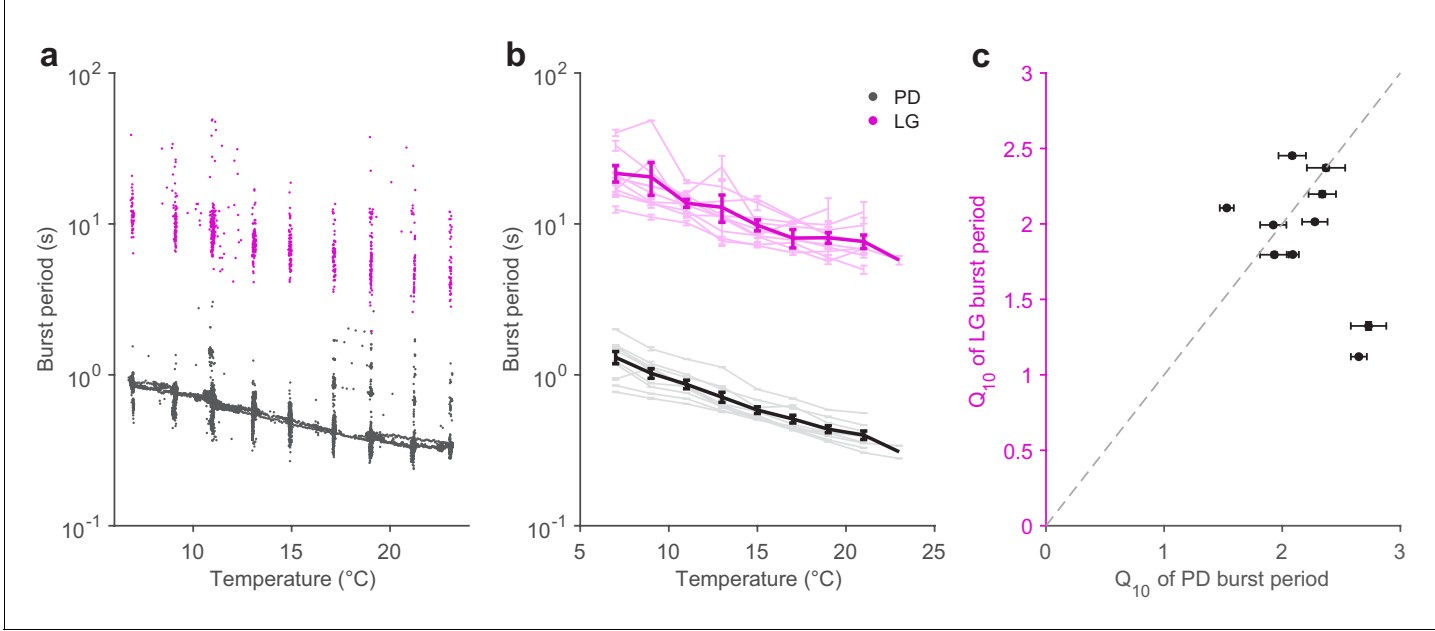

**Figure 4.** Pyloric and gastric mill periods are similarly temperature-sensitive. (**a**) Burst period of PD (black) and LG (magenta) in a single preparation as a function of temperature. Each dot corresponds to a single burst. (**b**) Burst period of PD and LG for 10 preparations. Each line is a single preparation. Mean of all preparations shown in thick lines. (**c**) The apparent $Q_{10}$ of the burst frequency of each preparation for the PD and LG neurons, representing the pyloric and gastric mill rhythms. The apparent $Q_{10}$s of the two rhythms are not significantly different (paired t-test, $p = 0.235$, $N = 10$, $t = 1.27$)).

The online version of this article includes the following figure supplement(s) for figure 4:

**Figure supplement 1.** Gastric mill rhythms slow down with time after stimulation.

**Figure supplement 2.** Evoked gastric mill rhythms persist when the pyloric pacemaker is suppressed.

(*Figure 4—figure supplement 1*) of the gastric mill rhythm with time since stimulation (*Beenhakker et al., 2004*; *Stein et al., 2007*).

A coordinated decrease in both gastric mill and pyloric cycle periods was observed in all 10 preparations (*Figure 4b*). We quantified the extent to which the cycle period increased with temperature using the $Q_{10}$ (Materials and methods). *Figure 4c* shows that most preparations (8 of 10) exhibited a cycle period $Q_{10}$ of ~2 for both the pyloric and gastric mill rhythms. Estimated $Q_{10}$s were not significantly different for LG and PD burst periods (paired t-test, $p = 0.235$, $N = 10$, $t = 1.27$), suggesting that the periods of both circuits were not differently influenced by increased temperature.

## Gastric mill rhythms can be evoked when the pyloric pacemaker is suppressed

At 11°C, the gastric mill half-center is capable of producing rhythmic activity even if the pyloric rhythm is turned off by hyperpolarization of the pacemaker neurons (*Bartos et al., 1999*). Therefore, we hypothesized that the gastric mill oscillator might be independently temperature-robust in the absence of pyloric input. To test this, we evoked gastric mill rhythms at 11°C, 15°C, 19°C, and 21°C with the pyloric pacemaker both active and inactive (Materials and methods).

*Figure 4—figure supplement 2a and b* show example recordings from a single preparation in which, at each temperature, we evoked a 'control' gastric rhythm (AB/PD active) followed by an evoked gastric rhythm with both PD neurons hyperpolarized (AB/PD inactive). *Figure 4—figure supplement 2c* shows that the gastric mill rhythm cycle period steadily decreased with increased temperature regardless of pyloric pacemaker activity. *Figure 4—figure supplement 2d* shows that in all nine preparations, the gastric mill rhythm was activated at 11°C and 15°C, whereas at 19°C and 21°C only a subset of preparations produced gastric mill rhythms at these elevated temperatures.

## Integer coupling is maintained across a physiological temperature range

Integer coupling between two interacting oscillators with different periods refers to the phenomenon in which the slower rhythm (gastric mill) tends to have a period that is an integer multiple of the period of the faster cycle (pyloric; *Nadim et al., 1998*). Integer coupling exists between the pyloric and gastric mill oscillators at 11°C (*Bartos et al., 1999*; *Hamood and Marder, 2015*). It remains an open question if this form of precise coordination is preserved across temperature, because both oscillators change their cycle periods with temperature.

Extracellular recordings of *lgn* and *pdn* at 7°C and 23°C from the same preparation are shown in *Figure 5a*. PD bursts tend to terminate shortly before the initiation of LG bursting at both temperatures (*Figure 5a*, dotted vertical lines indicate start of LG bursting). The ratio of LG to PD periods was close to an integer at both temperatures (~10 at 7°C and ~11 at 23°C).

To visualize integer coupling between the PD and LG across the entire dataset, we plotted the LG burst period as a function of the mean PD burst period during that LG burst (*Figure 5b*). Data tended to lie along lines with integer slope (gray lines), suggesting integer coupling between LG and PD across temperature. DG burst periods, which were similar to LG burst periods, did not tend to lie on lines with integer slope (*Figure 5c*). To quantify the degree of integer coupling, we computed the cumulative distribution of significands (value after the decimal point) of the ratios of gastric mill to pyloric periods (*Figure 5d*, Materials and methods). Perfect integer coupling would lead to a cumulative distribution indicated by the dotted line, while randomly chosen significands would lie along the diagonal. Cumulative distributions for LG were skewed towards perfect integer coupling, while distributions for DG were closer to the randomly distributed significands, suggesting that integer coupling was observed in LG but not DG.

To quantify how the degree of integer coupling varies across temperature, we measured the area between the cumulative distributions and the diagonal in *Figure 5d*. Perfect integer coupling would yield an area of 0.25. Areas for LG/PD significands were larger than areas for DG/PD significands at every temperature (*Figure 5e*), suggesting greater integer coupling. Areas for DG/PD significands overlapped with areas computed after shuffling gastric mill neuron (both LG and DG) and PD neuron cycle periods (gray lines, *Figure 5e*). To determine if measured integer coupling was significantly different from chance, we used a two-sample Kolmogorov-Smirnoff test (Materials and methods). LG and PD were significantly integer-coupled, at temperatures from 7°C to 21°C ($p < 10^{-5}$, $D > 0.24$), while DG and PD were not significantly coupled at any temperature ($p > 0.02$, $D < 0.16$). AB dictates the phase at which LG turns on because LG onset is timed with the peak of Int1 inhibition (*Nadim et al., 1998*). Because all previous studies of integer coupling were at ~11°C, it was not known if the process underlying integer coupling would shift with temperature increase. To visualize the effect of temperature on the ratio of PD to LG cycle period, we plotted the ratio of LG cycle period and PD cycle period as a function of temperature (*Figure 5f*). At low temperatures, when integer coupling was particularly strong (as seen from the horizontal banding), these ratios were distributed over a wide range from 5 to 25. Ratios of LG to PD cycle period were spread across a similar range of temperatures, even at higher temperatures when integer coupling was less pronounced ($p = 0.88$, $n = 3799$, PD cycles pooled across all preparations, Spearman test), suggesting that the mechanism of integer coupling was independent of the number of PD cycles to LG cycle period.

## LG preserves phase locking to PD across physiological temperatures

How do LG and PD maintain integer coupling in their burst periods across this temperature range? One possibility is via temperature-invariant phase locking between LG and PD. If LG bursts start at a particular phase in the PD cycle, then the burst periods of LG and PD must be integer-coupled (*Nadim et al., 1998*). We, therefore, set out to study the fine structure of the phase coupling between LG and PD across temperature.

To visualize the phase coupling between LG and PD, we plotted PD spikes aligned to LG burst starts (*Figure 6a*) normalizing time by the mean PD burst period in each row. Doing this aligns all PD spikes, suggesting that LG burst starts were an excellent predictor of PD phase. To quantify this effect across all preparations, we computed probabilities of LG burst starts across the temperature range tested. LG burst start probabilities were sharply peaked (*Figure 6b*) and were significantly different from a uniform distribution (*Figure 6b*, dashed line, $p < 4 \times 10^{-4}$, $7.5 < z < 45$, Rayleigh test for

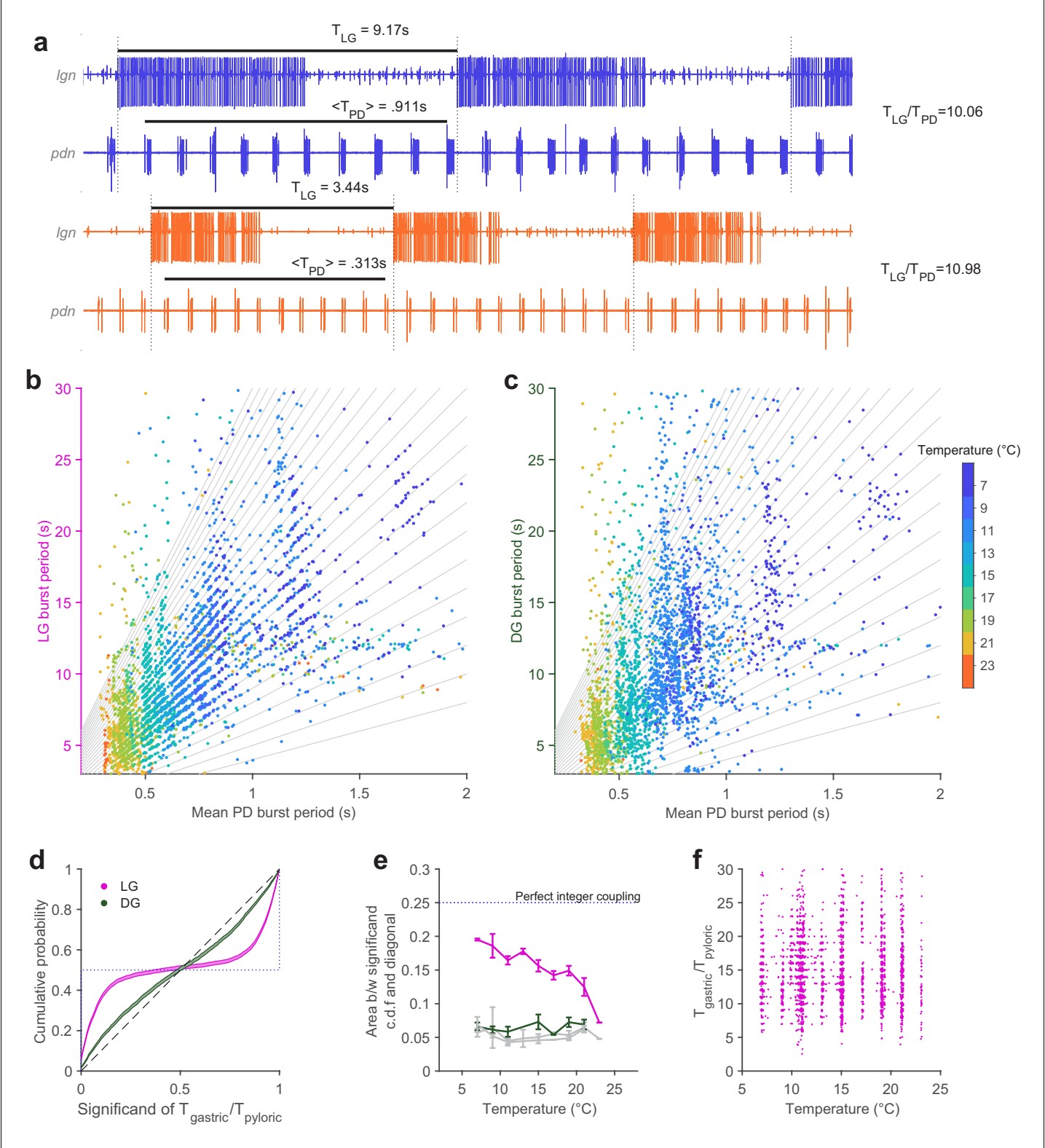

**Figure 5.** Integer coupling is maintained over physiological range. (**a**) Extracellular recordings of *pdn* show bursting of LG and PD at 7°C (blue) and 23°C (orange). Black bars indicate burst period of LG and mean burst period of PD. LG (**b**) and DG (**c**) burst period as a function of mean PD period. In (**b–c**), gray lines have integer slopes and are not fits. (**d**) Cumulative probability distributions (c.d.f.s) of significand of ratio of gastric to pyloric period for LG (magenta) and DG (green) for all temperatures. Shading indicates confidence interval estimates from bootstrapping the data. Dashed diagonal line indicates a uniform distribution, and the dotted line indicates perfect integer coupling. (**e**) Area between significand c.d.f.s and diagonal (a measure of

*Figure 5 continued on next page*

*Figure 5 continued*

how integer-coupled the circuits are) as a function of temperature for all preparations. Gray lines are computed from shuffled data. LG and PD were significantly integer coupled between 7°C and 21°C (p < 10^{-5}, D > 0.24, two-sample Kolmogorov-Smirnoff test), while DG and PD did not exhibit integer coupling. (f) Ratio of gastric to pyloric periods as a function of temperature. Apparent banding along the temperature axis is a consequence of the experimental protocol, where discrete temperature steps were used (Materials and methods). Ratios of LG to PD cycle period were n across temperature (p = 0.88, Spearman test).

non-uniformity; see Methods for details). From 7°C to 21°C, LG bursts started at a phase of ~0.5 in the PD cycle (*Figure 6b*, inset), and this phase offset did not significantly trend across this temperature range (p = 0.46, ρ = –0.31, Spearman test).

LG burst initiation being phase-locked to PD suggests that faster dynamics in LG, such as spiking, could also be affected by PD. Modulation of LG spiking by AB/PD has been reported via feedback from AB to MCN1 and CPN2 (*Blitz and Nusbaum, 2008*) and possibly Int1 feedback to CPN2 (*Blitz, 2017*). We measured the spike probability of LG conditional on PD phase across this

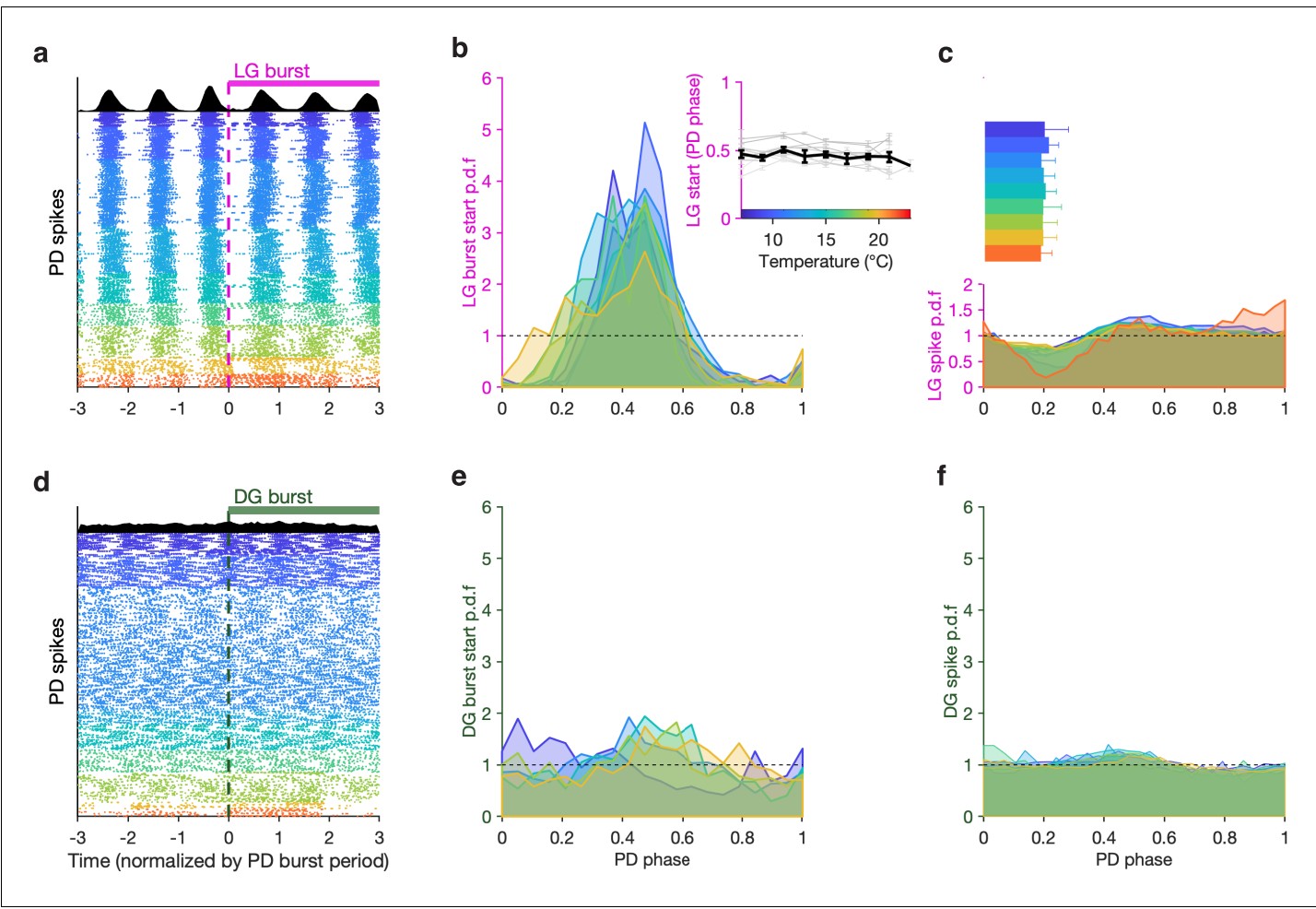

**Figure 6.** LG, but not DG, bursts and spikes are phase-locked to PD across temperatures. (a) PD spike rasters aligned to LG burst starts (dashed line at 0) in a single preparation. Each row corresponds to an LG burst start, and time is normalized by PD burst period. Black histograms at the top show spike probability over all rasters. (b) LG burst start probability as a function of PD phase. Inset shows mean LG burst start phase a function of temperature (gray lines: individual preparations, black line: mean across all preparations). LG burst start probabilities were significantly different from a uniform distribution (p < 4 × 10^{-4}, 7.5 < z < 45, Rayleigh test for non-uniformity). Phase offset between LG and PD was similar from 7°C to 21°C (inset, p = 0.46, Spearman test). (c) LG spike probability as a function of PD phase at various temperatures. Bars indicate PD spiking. (d–f) Similar to (a–c), but for DG. DG burst starts are not always significantly non-uniformly distributed, and the effect size is much smaller (p ∈ [0.001, 0.85], 0.15 < z < 13, Rayleigh test). Rasters in (a,d) are from the same preparation.

temperature range and found a pronounced decrease in the probability of spiking in LG at higher temperatures (*Figure 6c*). This decrease in LG spike probability occurred close to PD burst termination (*Figure 6c*, horizontal bars).

Although DG bursts on the same timescale as LG, it is not directly coupled to the pyloric oscillator (*Figure 1*) and does not show any signature of phase coupling with PD. DG bursts starts do not align with PD spikes (*Figure 6d*), DG burst starts are not sharply peaked in PD phase, and they are not significantly different from a uniform distribution for all preparations at all temperatures (*Figure 6e*, p ∈ [0.001, 0.85], 0.15 < z < 13, Rayleigh test). DG spike probability also does not show a noticeable decrease close to the end of PD bursting (*Figure 6f*, *Figure 6c*).

## Discussion

### Temperature-sensitive circuits maintain coordinated activity

Coupling between rhythmic circuits is common in nervous systems and is often functionally important, although the strength and the extent of inter-circuit coupling is varied (*Nadim et al., 1998*; *Bartos et al., 1999*; *Colgin, 2011*; *Gordon, 2011*; *Jacobson et al., 2013*; *Rojas-Líbano et al., 2014*; *Harris and Gordon, 2015*; *Karalis et al., 2016*; *Tamura et al., 2017*). Temperature is a global perturbation that influences the properties of all ion channels and synapses in neuronal circuits (*Zečević et al., 1985*; *Janssen, 1992*; *Xu and Robertson, 1996*; *Garrity et al., 2010*; *Tang et al., 2010*; *Kang et al., 2012*; *Robertson and Money, 2012*; *Marder et al., 2015*). Because each cellular and molecular process is likely to have a different temperature dependence, or $Q_{10}$, maintaining the function of complex physiological processes as temperature is changed is non-trivial (*Du et al., 2010*; *Caplan et al., 2014*; *Roemschied et al., 2014*; *O'Leary and Marder, 2016*; *Schleimer and Schreiber, 2016*). Most often, changes in temperature affect neuronal and circuit behavior (*Zecevic and Levitan, 1980*; *Prosser and Nelson, 1981*; *Long and Fee, 2008*; *Ramot et al., 2008*; *Garrity et al., 2010*; *Robertson and Money, 2012*). It then becomes critical to know whether the coupling between circuits is also altered (*Stiebler and Narins, 1990*), or whether the relative coordination between rhythmic behaviors can be maintained as temperature changes.

Here we show that two oscillator circuits can maintain coordinated output across a range of physiologically relevant temperatures. The cardiac ganglion, another circuit in the crab, which governs heartbeat, is also temperature-sensitive across a range of physiological temperature, again with a $Q_{10}$ of ~2 (*Kushinsky et al., 2019*). In the pyloric, gastric mill and cardiac circuits, rhythmicity is maintained up to a critical temperature, and this critical temperature approximates the upper range of native ambient temperatures (*Tang et al., 2010*; *Städele et al., 2015*; *Kushinsky et al., 2019*).

Coupling of the gastric mill and pyloric circuits is likely to be important for the crab. In the intact animal, muscle movements of the gastric mill enable ingested food to be chewed and subsequently filtered and pumped into the hindgut by the pylorus (*Marder and Bucher, 2007*). Coupling between these circuits coordinates the timing of chewing and filtering enabling proper flow of food through the mid-gut (*Meyrand et al., 1994*; *Diehl et al., 2013*). Furthermore, a subset of STG neurons participates in both rhythms, and therefore coupling between the pyloric and gastric mill circuits enables neurons with dual functions to maintain their appropriate participation in both rhythms (*Weimann et al., 1991*; *Weimann and Marder, 1994*; *Gutierrez et al., 2013*; *Blitz et al., 2019*).

### Oscillator circuits with different periods are similarly temperature-robust

Although the gastric mill and pyloric rhythms are similarly temperature-sensitive, they have an approximately 10-fold difference in cycle period. The pyloric rhythm is temperature-robust because the AB neuron maintains a relatively constant duty cycle across this range (*Tang et al., 2010*; *Rinberg et al., 2013*). Temperature robustness in the gastric mill rhythm arises from a different set of interdependent processes including projection neuron input, coordination of the half-center oscillator neurons LG and Int1, and feedback from LG to MCN1. The similarity between the apparent cycle period $Q_{10}$s of the pyloric and gastric mill rhythms was unexpected as there are so many different physiological processes generating the two rhythms. Moreover, a lock step variance in cycle period is not necessary in order to maintain coupling as the relative number of pyloric cycles between each gastric cycle can vary widely while circuits remain coupled (*Nadim et al., 1998*;

*Bartos et al., 1999*). Because the salient drivers of LG firing are sufficient projection neuron excitation and disinhibition from Int1 (*Bartos et al., 1999*; *DeLong et al., 2009*), the total number of pyloric cycles per gastric mill cycle can vary greatly. To this end, it would be interesting to know how increasing and decreasing pyloric cycle period via excitation of both PD neurons would affect coupling between the two rhythm generators across temperature.

## Spontaneous gastric mill rhythms were non-stationary and driven by unknown processes

As spontaneous gastric mill activity was often irregular, it was difficult to examine coupling between the two rhythm generating circuits. The observed irregularity could be due to weak activation of one or more descending pathways and is likely to be different across preparations, and potentially across temperatures. There are many different processes that drive gastric mill rhythms including release of hormonal peptides such as crustacean cardioactive peptide (CCAP; *Weimann et al., 1997*), FLRFamides (*Heinzel et al., 1993*; *Weimann et al., 1993*; *Christie et al., 2004*), pyrokinins (*Saideman et al., 2007b*; *Dickinson et al., 2015*), and Gly$^1$-SIFamide (*Blitz et al., 2019*). Synaptically released peptides *C. borealis* tachykinin related peptide (CabTRP Ia; *Blitz et al., 1999*; *Wood et al., 2000*), and *C. borealis* pyrokinin peptide I and II (CabPK; *Saideman et al., 2007a*; *Saideman et al., 2007b*) generate similar gastric mill rhythms, while proctolin (*Blitz et al., 1999*; *Stein et al., 2007*) excites all gastric mill motor neurons.

Stimulation of certain projection neuron such as MCN1 (contains CabTRP, proctolin and GABA; *Coleman and Nusbaum, 1994*; *Coleman et al., 1995*; *Bartos and Nusbaum, 1997*), MCN5 (contains Gly$^1$-SIFamide; *Norris et al., 1996*; *Blitz et al., 2019*), MCN7 (contains proctolin; *Blitz et al., 1999*), CPN2 (*Norris et al., 1994*), and IVN (the inferior ventricular neurons; contain FLRFamide and histamine; *Christie et al., 2004*), or stimulation of sensory pathways that activate groups of projection neurons (*Katz and Harris-Warrick, 1991*; *Beenhakker et al., 2004*; *Beenhakker and Nusbaum, 2004*; *Blitz et al., 2004*; *Beenhakker et al., 2007*; *White and Nusbaum, 2011*) generate gastric mill rhythms. Because there are so many ways gastric mill rhythms, and because each method of gastric mill rhythm generation could result in a gastric mill rhythm via a different set of mechanisms, they could have different temperature dependencies. Consequently, we chose to elicit gastric mill rhythms using a single, known mechanism, and we used exogenous stimulation that would reliably produce gastric mill rhythms, that were qualitatively similar to the spontaneous gastric mill rhythms we had previously observed.

A previous study showed that one gastric mill rhythm variant is easily disrupted by small temperature changes (*Städele et al., 2015*), while the one studied here maintains rhythmicity following a large increase in temperature. Although the functional significance of the diversity among gastric mill rhythms is unclear, the possibility remains that the assortment of gastric mill rhythm generator mechanisms bequeaths a robustness against environmental perturbation. Because there are other versions of the gastric mill not considered here, it will be interesting in the future to determine the extent to which they are temperature-robust and whether, in turn, they are coupled to the pyloric rhythm across temperature.

## Temperature compensation is driven by multiple mechanisms

Previous work has shown that neuromodulatory substances can extend the temperature range over which the gastric and pyloric rhythms maintain function (*Städele et al., 2015*; *Haddad and Marder, 2018*). In *C. elegans* neuromodulation enhances the robustness of thermosensory neurons and altering temperature-dependent behavior (*Beverly et al., 2011*). Therefore, it is possible that some modulators that act on the STG, not studied here, may further stabilize these rhythms, and some of the modulatory inputs to the STG may release transmitter in response to temperature stress for precisely this reason.

The gastric mill oscillator is activated by both chemical and electrical synapses. In crayfish, the rectifying electrical junction from the lateral giant fiber to the motor giant fiber has reliable synaptic propagation between 20°C and 30°C (*Heitler and Edwards, 1998*). In a model of this crayfish circuit, *Heitler and Edwards, 1998* found that if the electrical synapse had a $Q_{10}$ of 11, when most other measured properties of the circuit had $Q_{10}$s between 2 and 5, the model reliably reproduced physiological responses to temperature increase. In this system, at high temperatures, this increased drive

from the electrical synapse offsets the decrease in input resistance of the postsynaptic neuron. In the evoked gastric mill rhythms, CPN2 drives LG and GM via electrical synapses (*Figure 1c*), and our spontaneous rhythms, indicative of CPN2 activation, were observed up to 30°C. While we do not know the $Q_{10}$ of electrical synapses in the STG, several measured chemical synapses in the pyloric circuit have a $Q_{10}$ of ~2.5 (*Tang et al., 2010*).

Previous work showed that input resistance modestly decreases in both the Lateral Pyloric (LP) and LG neurons with increased temperature (*Tang et al., 2010*; *Städele et al., 2015*). The MCN1– LG electrical synapse may help offset the change in input resistance in the gastric circuit. In the pyloric circuit, the IPSP amplitude in the LP neuron is temperature-invariant ($Q_{10}$ ~1), but the IPSC amplitude increased ($Q_{10}$ ~2.5) across the same range of temperatures as those studied here (*Tang et al., 2010*). The increase in synaptic current may partially compensate for the increase in leak as temperature increases. This could aid circuits in maintaining rhythmicity across temperature. Furthermore, many STG neurons, including both LP and LG, exhibit post-inhibitory rebound (PIR). Both $I_A$ and $I_h$ (currents that promote bursting and PIR) exhibit temperature-dependent increases in activation rate and peak current ($Q_{10}$s ~ 2 to 4; *Tang et al., 2010*).

## Circuits are robust to global perturbation

Although it may not be uniformly true that all neuronal circuits maintain coordinated function across a wide range of temperatures in cold-blooded animals, here we demonstrate that for two coupled circuits, where coupling is necessary for proper function, coordination is maintained despite a global perturbation. Because each circuit's oscillations arise from a different set of circuit and cellular mechanisms, it is unlikely that the underlying processes endowing temperature compensation are the same. Therefore, it was not necessarily expected that compensation would be similar for both circuits.

The nervous systems of small homeotherms such as birds, rodents (*Chaffee and Roberts, 1971*; *Janssen, 1992*) and even human infants (*Mank et al., 2016*; *Barbi et al., 2017*) likely face similar challenges as a result of fluctuations in ambient temperature or in fever response to illness. Although they have homeostatic processes that regulate internal temperature shifts, these regulatory responses can be slow compared to temperature change (*Cheshire, 2016*). Therefore, it is likely that their nervous systems must be robust to some degree of temperature fluctuation to maintain function until a homeostatic process can intervene to restore internal temperature. While mouse brain neuron temperature sensitivities exist in a much narrower range than the crustaceans studied here (*Hori et al., 1999*), the temperature range we use is well within the range experienced by the crabs naturally.

In this study, we use temperature as a global perturbation. It is possible that coupled neural circuits are similarly robust to other global perturbations, so that coupling is maintained despite transient or chronic changes in ambient conditions. For example, the pyloric oscillator is known to be robust to large changes in pH (*Haley et al., 2018*) and to fluctuations in extracellular potassium (*He et al., 2020*). Collectively, these data indicate that the crustacean STG (and likely many other neural circuits) are capable of sustaining normal output when challenged with global perturbations. While the processes that confer robustness may be different for different circuits and perturbations, nervous systems have likely evolved to cope with a variety of environmental changes.

# Materials and methods

## Animals and dissection

Adult male Jonah crabs (*C. borealis*; *N* = 19 from Commercial Lobster [Boston, MA] and *N* = 9 from Moody's Seafood [Brunswick, ME]) were kept in tanks with circulating artificial sea water at 11°C for approximately 1 week prior to use. Preparations with spontaneous rhythms (*N* = 9) were from crabs caught in July 2013 and studied soon thereafter. Preparations in which rhythms were evoked (*N* = 10) were from crabs caught in July, September, and December of 2017. Preparations in which the pyloric pacemaker was inhibited (*N* = 9) were from crabs caught in August and September 2020. Thirty minutes prior to dissection, animals were chilled on ice. Dissections were carried out as previously described (*Gutierrez and Grashow, 2009*) and the dissected nervous systems were kept in chilled saline solution (440 mM NaCl, 11 mM KCl, 26 mM $MgCl_2$, 13 mM $CaCl_2$, 11 mM Trizma base,

5 mM maleic acid, pH 7.45 at 23°C) throughout the dissection. Each dissected stomatogastric nervous system was pinned out in a petri dish with a Sylgard-lined bottom and fresh saline was continuously superfused across the nervous system for the duration of the experiment (*Figure 1a*).

## Nerve recordings

Motor neuron action potentials originating in the STG (*Figure 1a*) were recorded extracellularly (*Figure 1b*) from the nerves via stainless steel pin electrodes. Stretches of the *lgn*, *dgn*, *mvn*, *lvn* and *pdn* were electrically isolated from the bath with Vaseline wells. Voltage signals were amplified using model 3500 amplifiers (A-M Systems) and digitized at 10 kHz using a Digidata 1440 A-D converter (Axon Instruments/Molecular Devices). Data were recorded using pClamp data acquisition software (Axon Instruments/Molecular Devices, version 10.5).

## Temperature control

Saline temperature was monitored continuously using a temperature controller (model CL-100, Warner Instruments) and altered during each experiment using an associated Peltier device and thermocouple (SC-20 and TA-29, Warner Instruments). Saline inflow to the nervous system was positioned within 1 cm of the STG so that the measured temperature at the point of inflow was approximately that of the ganglion somata.

## Gastric mill rhythm stimulation

We bilaterally stimulated the dorsal posterior esophageal nerves (*dpons*) to elicit a gastric mill rhythm (*Figure 1a, b*). Stainless steel pin electrodes were placed on either side of each *dpon* and were sealed to the nerve using Vaseline. Stimuli were delivered using a model 3800 stimulator (A-M Systems) via model 3820 stimulus isolation units (A-M Systems). Ten, six-second long episodic stimulus trains (0.6 Hz) were used to evoke gastric mill rhythms using a within-train stimulus rate of 15 Hz, as described in *Beenhakker and Nusbaum, 2004*. The *dpons* contain axons of the ventral cardiac neurons (VCNs) that activate the MCN1 and CPN2, resulting in a VCN-gastric mill rhythm that is often active for tens of minutes (*Beenhakker et al., 2004*; *Beenhakker and Nusbaum, 2004*; *Blitz and Nusbaum, 2008*). The duration of each stretch of gastric mill activity is not consistent either within an experiment at different temperatures or across experiments at the same temperature. Thus, the time between each gastric mill rhythm stimulation was different both between temperatures and across preparations.

## Defining gastric mill rhythm variants

While evoked gastric mill rhythms are distinguished by the method used to drive them, in the case of spontaneous gastric mill rhythms, the phase relationships and timing of the gastric mill neurons are used to classify them (*White and Nusbaum, 2011*).

Recordings from the *mvn* show the Inferior Cardiac (IC) neuron and ventricular dilator (VD) neuron (*Figure 1b*). Although, in the absence of a gastric mill rhythm, the IC and VD neurons are typically active in time with the LP neuron and the Pyloric neurons (PY$_5$; there are approximately five PY neurons in each preparation), respectively, the initiation of a gastric mill rhythm will often shift the IC and VD neurons into gastric mill timing as shown in *Figure 1b*. Because the timing of the IC and VD neurons are state-dependent, they can aid in classifying spontaneous gastric mill rhythms.

In this study, spontaneous gastric mill rhythms resemble VCN-rhythms as IC and VD activity is restricted to the LG interburst interval, which is known to occur because of CPN2 and LG inhibition of the IC and VD neurons, respectively. Additionally, the LG bursts in these spontaneous rhythms did not exhibit any obvious interruptions, as is the case for another gastric mill rhythm that is similar to a VCN-rhythm, save for this difference (*White and Nusbaum, 2011*).

## Spontaneous and evoked gastric mill rhythm temperature ranges

In one set of experiments ($N = 9$), spontaneous gastric mill rhythms were observed. Here, temperature was increased from 11°C to 33°C in 2°C increments, and a given temperature step was held for at least 2 minutes. Time between steps was ~2 minutes.

For experiments in which gastric mill rhythms were evoked ($N = 10$), the temperature was increased in either 2°C or 4°C steps starting at 7°C until a gastric mill rhythm could no longer be

evoked (typically 21°C or 23°C). Each temperature step was held for the duration of evoked gastric mill activity, and only after the epoch of the gastric mill rhythm terminated was the temperature advanced to the next step. The time between temperature steps was ~2 minutes.

The VCN-gastric mill rhythms were difficult to generate consistently above 23°C, and in some cases only persisted for several cycles. In some preparations, 19°C was the warmest temperature that a gastric mill rhythm could be evoked. Because the stimulation paradigm only activates a subset of projection neurons in the CoGs, these manipulations may fail to characterize the full extent to which the gastric mill circuit can maintain rhythmicity across temperature. Within the temperature range in which they were readily evoked, gastric mill rhythms lasted for tens of minutes.

### Evoked rhythms with AB/PD hyperpolarized

To study the effect of temperature on VCN-gastric mill rhythm alone, in a separate set of experiments, gastric mill rhythms were evoked twice at 11°C, 15°C, 19°C, and 21°C with the pyloric pacemaker both active and inactive at each temperature step ($N = 9$). To inactivate the pyloric pacemaker, each PD neuron was impaled with a single bridge balanced sharp electrode in a current clamp configuration. PD neurons were identified by briefly hyperpolarizing the membrane potential and monitoring the number of spikes recorded extracellularly on the *pdn*. Action potentials from both PD neurons can be monitored on a single *pdn*. Because the electrical coupling between the AB and PD neurons is strong (*Maynard and Selverston, 1975*; *Shruti et al., 2014*), it is not necessary to hyperpolarize all three neurons. PD neurons were targeted because of their relatively large size compared to the AB neuron. Because neurons swell with increased temperature (*Tang et al., 2010*), it can be difficult to maintain intracellular recordings, especially of small neurons, and not damage the cells.

### Data analysis

#### Spike identification and sorting

To reliably identify spikes from extracellular traces, we wrote spike sorting software that uses supervised machine learning to learn the shape of spikes from a small manually labeled dataset. This software is freely available at https://github.com/sg-s/crabsort (*Srinivas, 2021*). PD, LG and DG neuron spikes were identified on the *pdn*, *lgn* and *dgn* nerves recorded extracellularly. In a small subset of the full data, spikes from these neurons were identified using thresholding, dimensionality reduction of the full spike shape, and manual labeling. Using this subset, we trained a fully connected neural network to classify putative identified spikes in new data. The accuracy of the classification was constantly monitored during sorting, and new annotations were used to continuously re-train the network using an active learning framework. Classification accuracy always exceeded 98%.

After the neural network classified putative spikes in a new dataset, to ensure high-quality spike annotation: (1) we manually inspected the data in 2-minute chunks to ensure that no artifacts were accidentally labeled as spikes; (2) we used tools built in to the spike sorting software to visually inspect, at high-temporal resolution, spikes that the neural network reported with low confidence; and (3) we used other built-in tools to inspect spikes that differed statistically from other spikes in its class. These steps let us generate high-quality spike annotations for PD, LG, and DG, even when these neurons were partially obscured by other units or noise.

#### Detection of bursts and measurement of burst metrics

Bursts in PD, LG, or DG were defined using inter-spike intervals (ISIs). ISIs longer than 1 second on LG or DG were defined to be an inter-burst interval and were used to annotate the start of a burst. Spikes were considered to be part of a burst only if there were at least $n$ spikes per burst, with $n = 5$ for LG and DG and $n = 2$ for PD.

This method allowed us to identify all burst starts, burst stops, and therefore burst durations and burst periods for every burst. We computed the duty cycle on a cycle-by-cycle basis by dividing the burst duration by the burst period.

#### Estimation and comparison of $Q_{10}$s (*Figure 4*)

The $Q_{10}$ of a process is defined as the factor by which it changes over a 10° increment in temperature. We computed the $Q_{10}$ of PD and LG burst periods at every temperature using.

$$Q_{10} = \left\langle \left( \frac{f_T}{f_{11}} \right)^{\frac{10}{T-11}} \right\rangle$$ where $f_T$ is the mean burst frequency at temperature T and $f_{11}$ is the mean burst frequency at 11°C (the reference temperature).

To compare the $Q_{10}$s of the pyloric and gastric rhythms, we used a paired t-test and compared the mean $Q_{10}$s averaged across each preparation. Each animal thus contributed equally and we included all 10 preparations. $t$ represents the test statistic.

## Integer coupling analysis (*Figure 5*)

To measure the integer coupling between PD burst periods and LG or DG burst periods (*Figure 5b, c*), we computed the mean burst period of all PD bursts within one LG/DG burst period. Thus, every dot corresponds to one LG burst and all the PD bursts that occur within that LG burst period. LG/DG burst periods are plotted on the vertical axis and the mean PD burst period is plotted on the horizontal axis.

To determine if these burst periods are significantly integer-coupled (*Figure 5d*), we computed the significand (defined here as the value after the decimal point) of the ratio of the ordinate to the abscissa for each dot in *Figure 5b, c*. For example, if an LG burst had a period of 10.1 seconds, and the mean PD period during that LG burst was 1 second, the significand would be 0.1. If the LG burst period was 9.9 seconds, the significand would be 0.9. We sorted all significands for LG and DG and bootstrapped the data 1000 times to estimate confidence intervals of cumulative density functions of the significands. Shading in *Figure 5d* indicates these confidence intervals. Curves that deviated substantially from the diagonal (the null distribution) corresponded to significant integer coupling between the gastric mill and pyloric burst periods. To determine if any set of significands was significantly different from the null distribution, we performed a two-sample Kolmogorov-Smirnoff test comparing the significands from the ratios of LG/DG to PD burst periods and significands from ratios of shuffled copies of LG/DG and PD burst periods. The reason we used a two-sample Kolmogorov-Smirnoff test, and not a one-sample test that compares to a uniform distribution directly, is that we cannot expect significands to be distributed exactly uniformly, even for randomly distributed data. We observed through numerical calculations that for certain combinations of numerator (gastric cycle period) and denominator (pyloric cycle period) ranges, even randomly sampled cycle periods can lead to non-uniform significand distributions. The threshold for significance was divided by the number of comparisons (9 temperature steps) to correct for multiple comparisons. Data in *Figure 5d* are pooled across all preparations and across all temperatures. Statistical tests were performed grouping by temperature, but pooling across all preparations. $D$ indicates the Kolmogorov-Smirnoff test statistic, which is a measure of the distance between the empirical c.d.f and the model c.d.f.

The dotted and dashed lines in *Figure 5d* indicate two extremes of the significand distribution. If the gastric and pyloric burst periods were perfectly integer-coupled, then every significand of every ratio of periods would be 0 (or 1). Thus, the cumulative probability of such an idealized distribution would be a step-like function shown by the dotted line. On the other extreme, if the ratios of gastric to pyloric burst periods were such that the significands were randomly and uniformly distributed across the unit line (suggesting no integer coupling whatsoever), then the cumulative probability of such a distribution would be close to a diagonal line with unit slope, shown by the dashed line.

These two extremes allow us to define a yardstick to measure the degree of integer coupling for any observed spike train. We observe that all possible cumulative distributions are bounded by the two extremal distributions (dotted line and dashed line). Because the area of the square in *Figure 5d* is 1, simple geometry tells us that the area bounded by the maximal distributions is 0.25. We therefore quantified the degree of integer coupling of the two burst periods (*Figure 5e*) by measuring the area between the null distribution (dashed diagonal line) and the significand c.d.f.

To determine if the degree of integer coupling is significantly different from random at various temperatures (*Figure 5e*), we performed two-sample K-S tests as described above and divided the significance level by the number of comparisons (9). Our significance threshold was thus .05/9 = 0.0056.

### Generation of burst-aligned rasters of PD (*Figure 6a, d*)

To visualize the interaction between pyloric and gastric mill rhythms, we generated burst-aligned rasters of the PD neuron activity in which we aligned PD rasters to the burst start in a gastric mill-timed neuron (LG or DG). For every burst start in LG/DG, we plotted the raster of PD spikes centered around that burst start, showing spikes 3 s before the burst start in LG/DG and 3 s after.

### Estimation of gastric mill spike and burst start probability in PD phase (*Figure 6*)

To estimate when the LG or DG neurons start bursting in the PD cycle (*Figure 6*), we first identified the burst start times of LG and DG. At these points, we estimated the PD phase by dividing the time since the start of the previous PD burst by the burst period of that PD cycle. This method of phase estimation assumes that the phase increases uniformly and linearly over a PD cycle, following earlier work (*Prinz et al., 2004*; *Bucher et al., 2005*). We used a similar procedure to estimate gastric mill spike probability in PD phase.

### Analysis of distribution of LG/DG burst starts in PD phase (*Figure 6b, e*)

LG burst starts in PD time follow circular statistics, since a LG burst start occurring at a PD phase of 1 is the same as a LG burst start at a PD phase of 0. We therefore used a circular statistics toolbox by *Berens, 2009* (http://www.eye-tuebingen.de/berenslab/technology-development/) to analyze this data. Means and standard deviations of LG and DG burst starts (*Figure 6b* inset and *Figure 6e* inset) were computed using this circular statistics toolbox. To test if LG and DG burst starts were significantly non-uniformly distributed, we used the Rayleigh test for non-uniformity. $z$ is the Rayleigh test statistic. The distribution of burst starts of LG or DG in PD time for each preparation, at each temperature was tested for non-uniformity individually.

The number of LG/DG burst starts was not the same for every preparation at every temperature step. For this reason, and to avoid time-dependent confounds, we analyzed the first 40 burst starts for each preparation at each temperature step. Preparation and temperature combinations where there were <40 burst starts (~35% of data) were excluded. A total of n = 3816 LG burst starts and n = 3228 DG burst starts across nine temperature points and 10 preparations are plotted in *Figure 6b,d*.

### Data and code availability

Scripts to reproduce all figures in this paper are available at https://github.com/marderlab/gastric; *Powell, 2021*; copy archived at swh:1:rev:fa07d36a7e57fc7787a9a8e2fed72300b1a8e6e2. Raw data and spike annotations can be downloaded from https://zenodo.org/record/3924718.

## Acknowledgements

We thank Dr. Michael P Nusbaum for insights relevant to the operation of the gastric mill circuit.

## Additional information

### Funding

| Funder | Grant reference number | Author |
|---|---|---|
| National Institute of Neurological Disorders and Stroke | R35 NS 097343 | Eve Marder |
| National Institute of Neurological Disorders and Stroke | T32 07292 | Srinivas Gorur-Shandilya |

The funders had no role in study design, data collection and interpretation, or the decision to submit the work for publication.

## Author contributions
Daniel Powell, Conceptualization, Data curation, Investigation, Visualization, Methodology, Writing - original draft, Writing - review and editing; Sara A Haddad, Conceptualization, Data curation, Investigation, Visualization, Writing - review and editing; Srinivas Gorur-Shandilya, Conceptualization, Data curation, Software, Formal analysis, Visualization, Writing - original draft, Writing - review and editing; Eve Marder, Conceptualization, Supervision, Funding acquisition, Validation, Project administration, Writing - review and editing

## Author ORCIDs
Daniel Powell  https://orcid.org/0000-0002-9210-2201
Sara A Haddad  https://orcid.org/0000-0003-0807-0823
Srinivas Gorur-Shandilya  https://orcid.org/0000-0002-7429-457X
Eve Marder  https://orcid.org/0000-0001-9632-5448

## Decision letter and Author response
Decision letter https://doi.org/10.7554/eLife.60454.sa1
Author response https://doi.org/10.7554/eLife.60454.sa2

# Additional files

## Supplementary files
• Transparent reporting form

## Data availability
All electrophysiological data and analysis code has been uploaded to a publicly available data base: Scripts to reproduce all figures in this paper are available at https://github.com/marderlab/gastric copy archived at https://archive.softwareheritage.org/swh:1:rev:fa07d36a7e57fc7787a9a8e2-fed72300b1a8e6e2/. Raw data and spike annotations can be downloaded from https://zenodo.org/record/3924718.

The following dataset was generated:

| Author(s) | Year | Dataset title | Dataset URL | Database and Identifier |
| --- | --- | --- | --- | --- |
| Powell D, Haddad SA, Gorur-Shandilya S, Marder E | 2020 | Coupling between fast and slow oscillator circuits in Cancer borealis is temperature compensated | https://doi.org/10.5281/zenodo.3924718 | Zenodo, 10.5281/zenodo.3924718 |

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
