## [Decision Letter]

**Acceptance summary:**

The new experiments demonstrate that the gastric mill rhythm is maintained and displays temperature sensitivity even when the two neurons of the pyloric pacemaker group are hyperpolarized, thus removing the interactions with the pyloric circuit (Figure 4—figure supplement 2). In addition, the authors now illustrate the evolution of the burst period as a function of time (Figure 4—figure supplement 1), and these data nicely explain the degree of variability seen in the burst periods plotted in Figure 4. Considering the second set of requested experiments, the authors provide arguments explaining why they could not realistically carry out such difficult experiments in a short time window.

**Decision letter after peer review:**

Thank you for submitting your article "Coupling between fast and slow oscillator circuits in *Cancer borealis* is temperature compensated" for consideration by *eLife*. Your article has been reviewed by three peer reviewers, and the evaluation has been overseen by Ronald Calabrese as the Senior Editor. The following individuals involved in review of your submission have agreed to reveal their identity: Dimitri Ryczko (Reviewer #1); Muriel Thoby-Brisson (Reviewer #2); Farzan Nadim (Reviewer #3).

The reviewers have discussed the reviews with one another and the Reviewing Editor has drafted this decision to help you prepare a revised submission.

As the editors have judged that your manuscript is of interest, but as described below that additional experiments are required before it is published, we would like to draw your attention to changes in our revision policy that we have made in response to COVID-19 (https://elifesciences.org/articles/57162). First, because many researchers have temporarily lost access to the labs, we will give authors as much time as they need to submit revised manuscripts.

Summary:

This study addresses an important question for the physiology of coupled oscillatory neuronal networks operating under a wide range of temperatures. The stomatogastric system of the crab *Cancer borealis* contains the fast (~1Hz) pyloric network and the slow (~0.1 Hz) gastric mill network. The two generated rhythms are coordinated so that there is a given number of pyloric cycles per gastric cycle. Powell and colleagues show that upon stimulation of a neuromodulatory pathway, these coupled oscillatory circuits exhibit reproducible bouts of activity and maintain their coordination over a wide range of temperatures. The authors show that the gastric Lateral Gastric motor neuron (LG) is phase-locked with the Pyloric Dilatator neuron (PD), suggesting these neurons may be involved in coordination robustness.

Essential revisions:

While a large amount of raw data is nicely illustrated and the article is well written, all reviewers pointed out that the study is too descriptive and would benefit from using additional manipulations to reveal some of the mechanisms involved in the coordination of these oscillatory networks. In order to provide a mechanistic and more in depth analysis of the coupling, two issues should be addressed in the study. These two points could be potentially examined in the same set of experiments.

First, The intrinsic temperature sensitivity of the gastric rhythm (disconnected from the pyloric one) should be investigated:

While the intrinsic temperature sensitivity of the pyloric rhythm has been nicely investigated in previous publications, this property has been less studied for the gastric rhythm in the absence of the pyloric rhythm. The temperature sensitivity of the gastric mill rhythm should be examined independent of the pyloric rhythm, for example by silencing the pyloric pacemakers with inhibition or photoablation.

Second, it is important to know whether the robustness of the coupling persists similarly at different coupling strengths Because the AB synapse to the gastric mill half center oscillator is the essential coupling point, it would be important to manipulate the strength of this synapse (e.g. as it was done in Bartos et al., 1999) to see if its influence on the gastric mill period and the coupling remains the same across different temperatures.

Reviewer #1:

Powell and colleagues measured coordination robustness between pyloric and gastric rhythms in in vitro preparations of *Cancer borealis* exposed to temperature variations (7-23°C). Using extracellular recordings, they show that spontaneous rhythms are not stable, likely resulting from multiple physiological processes that are difficult to monitor. Therefore, they rather used bouts of activity reproducibly evoked by stimulation of a neuromodulatory pathway. Cold temperatures slowed down rhythms, warm temperatures accelerated rhythms in a similar manner. Rhythm coordination between pyloric and gastric rhythms was stable across temperatures despite variations in rhythm frequency. This suggested that rhythmogenic neurons coordinate their activity across temperatures. Using single cell electrophysiology, they found that the gastric Lateral Gastric motor neuron (LG) was phase-locked with the Pyloric Dilatator neuron (PD). Such substrate is likely involved in coordination robustness.

The originality of the study is that the authors focused on the coordination of pyloric (1 Hz) and gastric (0.1 Hz) networks. A large quantity of raw data is beautifully illustrated. Data analysis is sophisticated and convincingly supports the interpretations on the authors. The text is exquisitely written in a clear style and pleasant to read. A weakness of the study is maybe that it is a bit descriptive and lacks some mechanistic insights. In my view, the study contains the first experiments of a potentially exceptionally interesting study, but mechanistic insights would be needed to publish it in *eLife*. To further strengthen the relevance of the study, I would suggest one of the three options below to further uncover the mechanisms underlying the effects described. The authors will know what the most realistic and relevant experiments are.

1) Could the authors design causality-based experiments to identify which neuron is responsible for the coordination of the rhythms at different temperatures? There are many interconnected neurons in Figure 1C. Even if LG is phase locked to PD, is it possible that another neuron drives PD and LG? If PD controls LG, would it be relevant if the authors reversibly switched off PD (e.g. with tonic hyperpolarisation) and see the effect on gastric rhythm frequency at various temperatures?

2) Could the authors identify using pharmacological tools whether distinct neuromodulatory substance influence coordination robustness over specific ranges of temperature, but not in others? It seems that Stadele, Heigele and Stein, 2015, used a different way to evoke the rhythm, and their gastric rhythm crashed at lower temperatures (13°C) than in the present study (27°C). Do the authors think that the different stimulation approaches used in the two studies could involve different neuromodulatory substances, which would result in different robustness profiles?

3) Do the same intrinsic properties or synaptic connections underlie coordination robustness across temperatures? Modeling suggests that different conductances are involved in a temperature-dependent manner (Alonso and Marder 2020 e*Life* 9:e55470.2020). Is it possible for the authors to experimentally deactivate specific conductances using dynamic clamp in LG or PD or with pharmacological tools and determine whether this would reversibly disrupts the coordination between pyloric and gastric networks in some specific temperature ranges but not in others?

Reviewer #2:

In the present paper, Powell and colleagues investigated how coupled oscillatory circuits maintain their coordination over a wide range of temperature. To do so they used the stomatogastric system of the crab *Cancer borealis* that contains the fast (1Hz) pyloric network and the slow (0.1 Hz) gastric mill network. The two generated rhythms are coordinated such that there are an integer number of pyloric cycles per gastric cycle. Both rhythms exhibit temperature-induced frequency changes, but their coordination is well maintained even at high temperature. Therefore, this study shows that the relative coordination between rhythmic circuits can be maintained as temperature changes, thus ensuring appropriate physiological functions even under global perturbations.

This study, that uses a fantastic model for investigating neural networks in general, addresses an important physiological question. However, I have a few concerns that could be probably clarified with some additional explanations in the text:

– While the intrinsic temperature sensitivity of the pyloric rhythm has been nicely investigated in some previous excellent publications (most done by the authors), that of the gastric rhythm is less well known. Stadele has shown that increasing the temperature leads to a breakdown of the gastric rhythm that can be rescued by modulatory afferences. What do we know about the temperature sensitivity of the afferent neurons that are stimulated to trigger the gastric rhythm here? Is there the possibility that what is observed also includes an effect of the temperature changes on these neurons (MCN1 function for example) or that the gastric temperature sensitivity described here reflects in fact that of the afferences?

– All experiments were performed in conditions in which the gastric rhythm is triggered by stimulation of the two dorsal posterior esophageal nerves (*dpons*) that contain axons of modulatory afferent neurons. However stimulating these nerves also modulates the pyloric network that is also a target of those afferences (as stipulated in the subsection “Gastric Mill Rhythm Stimulation”). Isn't this a bias in the experiments and their interpretations? Also, because as schematically represented in Figure 1, the pyloric pacemaker neuron AB has direct connections with Int1 gastric neuron that is itself connected the LG gastric neuron, the simplest interpretation of the experiments would be that this connection is preserved and remains efficient even under high temperature. Is it finally one of the conclusions of the paper?

– In the same vain, the sensitivity to temperature changes of the gastric rhythm has been studied here but with the pyloric network, being itself intrinsically sensitive to temperature changes, still active (Figure 3 and related text). What do we know on the intrinsic temperature sensitivity of the gastric rhythm when elicited by *dpons* stimulation but isolated from the pyloric network (AB neuron killed for example)?

– Data presented here show that coordination between PD and LG neurons is preserved after temperature increase, but that this is not the case between PD and DG neuron that shows no phase-coupling at high temperature (Figure 6). The PD neurons is used here as an indicator of the pyloric rhythm while the LG neurons indicates the gastric rhythm. Then what would be the conclusions of the authors if the DG neuron would have been used as the gastric rhythm indicator? How do you conciliate everything together?

Reviewer #3:

The authors examine the robustness of coupling of distinct oscillatory circuits of different frequencies across a range of temperatures. The two circuits have different means of generating oscillations and could therefore, potentially, be impacted to different degrees by temperature perturbations. Across all temperatures tested the two distinct rhythms increased their frequency but remained coordinated. The coordination was in the form of the previously-described integer-coupling where the cycle period of the slow rhythm was an integer multiple of that of the fast one. This is due to the fact that the slower rhythm was most likely to start at a given phase within the faster oscillation cycle. The temperature robustness of this coupling is an interesting and important result and the description and analysis are both well done.

The main finding of the paper is that a previously-described integer-coupling between two rhythms remains more or less intact across temperature variations. It is a nice descriptive finding, but rather disappointing in that there is so much more that could have been done rather easily that would have given much more depth to this finding. Most obviously, because it is known that the source of the coupling is the inhibitory synapse from the pyloric pacemaker to the gastric mill half-center, it is quite important to know how the strength of this synapse affects the interaction at different temperatures. That is, to expand what Bartos et al., 1999, did across a range of temperatures. Short of that, it would have been nice at least to perturb the cycle period of the pyloric rhythm and see whether the interaction would remain robust across temperature despite changes in cycle period.

While the study convinces the reader that integer coupling between pyloric and evoked gastric rhythms is robust to temperature changes, it does not attempt to explore the origin of this robustness, e.g. by using different methods to activate the gastric rhythm or even testing if integer coupling is present with spontaneous gastric rhythms (which they do analyze for another purpose).

---

## [Author Response]

Essential revisions:While a large amount of raw data is nicely illustrated and the article is well written, all reviewers pointed out that the study is too descriptive and would benefit from using additional manipulations to reveal some of the mechanisms involved in the coordination of these oscillatory networks. In order to provide a mechanistic and more in depth analysis of the coupling, two issues should be addressed in the study. These two points could be potentially examined in the same set of experiments.First, The intrinsic temperature sensitivity of the gastric rhythm (disconnected from the pyloric one) should be investigated:While the intrinsic temperature sensitivity of the pyloric rhythm has been nicely investigated in previous publications, this property has been less studied for the gastric rhythm in the absence of the pyloric rhythm. The temperature sensitivity of the gastric mill rhythm should be examined independent of the pyloric rhythm, for example by silencing the pyloric pacemakers with inhibition or photoablation.

Now included: Figure 4—figure supplement 2; Results.

Second, it is important to know whether the robustness of the coupling persists similarly at different coupling strengths Because the AB synapse to the gastric mill half center oscillator is the essential coupling point, it would be important to manipulate the strength of this synapse (e.g. as it was done in Bartos et al., 1999) to see if its influence on the gastric mill period and the coupling remains the same across different temperatures.

This is an exceptionally difficult experiment, as it is very difficult to maintain intracellular recordings across temperature, and most prior studies repositioned the electrodes repeatedly (Tang et al., 2010; 2012), or only studied a small piece of the temperature range intracellularly (Rinberg et al; Ratliff, in review). Given that the AB neuron is small and fragile, and given that in the Bartos paper, at one temperature, the entire experiment was only achieved in a very small number of experiments, and given that any damage to the AB neuron would potentially invalidate the results, we think that producing a believable data set of this form would be very unrealistic.

Reviewer #1:[…] A weakness of the study is maybe that it is a bit descriptive and lacks some mechanistic insights. In my view, the study contains the first experiments of a potentially exceptionally interesting study, but mechanistic insights would be needed to publish it in eLife. To further strengthen the relevance of the study, I would suggest one of the three options below to further uncover the mechanisms underlying the effects described. The authors will know what the most realistic and relevant experiments are.1) Could the authors design causality-based experiments to identify which neuron is responsible for the coordination of the rhythms at different temperatures? There are many interconnected neurons in Figure 1C. Even if LG is phase locked to PD, is it possible that another neuron drives PD and LG? If PD controls LG, would it be relevant if the authors reversibly switched off PD (e.g. with tonic hyperpolarisation) and see the effect on gastric rhythm frequency at various temperatures?

To address this, in a new data set, we hyperpolarized the pyloric pacemaker to assess the temperature dependence of the gastric mill rhythm in the absence of AB input to the gastric mill circuit. Results, Figure 4—figure supplement 2.

2) Could the authors identify using pharmacological tools whether distinct neuromodulatory substance influence coordination robustness over specific ranges of temperature, but not in others? It seems that Stadele, Heigele and Stein, 2015, used a different way to evoke the rhythm, and their gastric rhythm crashed at lower temperatures (13°) than in the present study (27°). Do the authors think that the different stimulation approaches used in the two studies could involve different neuromodulatory substances, which would result in different robustness profiles?

While these experiments would add value to the data of the paper, these experiments involve recording from both the AB and Int1 neurons at varying temperatures. Recording from these neurons is difficult (due to their small size) and because neurons swell with increased temperature, maintaining these recordings at elevated temperatures is near impossible.

3) Do the same intrinsic properties or synaptic connections underlie coordination robustness across temperatures? Modeling suggests that different conductances are involved in a temperature-dependent manner (Alonso and Marder 2020 eLife 9:e55470.2020). Is it possible for the authors to experimentally deactivate specific conductances using dynamic clamp in LG or PD or with pharmacological tools and determine whether this would reversibly disrupts the coordination between pyloric and gastric networks in some specific temperature ranges but not in others?

We appreciate the reviewer's interest in this topic, but the experiments suggested would not produce a sufficient answer to what the reviewer is looking for. Eliminating a specific current would likely produce noticeable affects but it would not determine that current is responsible for the coupling/coordination/stability/robustness of the rhythms. Mimicking the absence or reduction of an intrinsic current in a specific cell would require first measuring that current in a specific cell across the temperature range and in the presence of blockers of other currents. These manipulations would interfere with subsequent results, and only be applicable to that cell (Schulz et al., 2006, 2007). It is also not possible to specifically block many of the modulatory currents pharmacologically. Lastly, due to the fact that we believe circuit robustness is a product of many possible solutions in conductance space which vary across individuals (Alonso and Marder 2020), which the reviewer points out, it is not experimentally possible to measure multiple conductances across multiple cell types and animals and it is likely that the answer is different across individuals.

Reviewer #2:[…] This study, that uses a fantastic model for investigating neural networks in general, addresses an important physiological question. However, I have a few concerns that could be probably clarified with some additional explanations in the text:– While the intrinsic temperature sensitivity of the pyloric rhythm has been nicely investigated in some previous excellent publications (most done by the authors), that of the gastric rhythm is less well known. Stadele has shown that increasing the temperature leads to a breakdown of the gastric rhythm that can be rescued by modulatory afferences. What do we know about the temperature sensitivity of the afferent neurons that are stimulated to trigger the gastric rhythm here? Is there the possibility that what is observed also includes an effect of the temperature changes on these neurons (MCN1 function for example) or that the gastric temperature sensitivity described here reflects in fact that of the afferences?

Stadele, Heigele and Stein, 2015 shows that MCN1 to LG EPSP amplitude decreases with temperature increase but that the loss of LG bursting capability is due to a temperature dependent increase in LG neuron leak (and can be occluded by injecting leak current with the dynamic clamp) rather than a loss of input from MCN1. While there are data showing reliable spike propagation across this temperature range for motor neurons in this nervous system (DeMaegd and Stein 2020), the only way to assess the temperature dependence of spike propagation for MCN1 or CPN2 would be to record from their axon terminals proximal to the STG (Coleman and Nusbaum, 1994). However, much like recording from AB and Int1, this would be exceedingly difficult to do across temperature, and would likely take many experiments to even collect a small dataset.

– All experiments were performed in conditions in which the gastric rhythm is triggered by stimulation of the two dorsal posterior esophageal nerves (dpons) that contain axons of modulatory afferent neurons. However stimulating these nerves also modulates the pyloric network that is also a target of those afferences (as stipulated in the subsection “Gastric Mill Rhythm Stimulation”). Isn't this a bias in the experiments and their interpretations?

Stimulating the *dpons* does drive slight difference in the pyloric rhythm but often for only a few cycles, and this does not affect the relative timing between the circuit generating neurons. This information is found in Beenhacker and Nusbaum, 2004. When calculating phase delays between LG and PD, the initial few cycles of each gastric mill rhythm were excluded for this very reason.

Also, because as schematically represented in Figure 1, the pyloric pacemaker neuron AB has direct connections with Int1 gastric neuron that is itself connected the LG gastric neuron, the simplest interpretation of the experiments would be that this connection is preserved and remains efficient even under high temperature. Is it finally one of the conclusions of the paper?

That is the most parsimonious explanation and what we believe to be taking place, however we cannot prove that with any of the experiments done here. These experiments would be extremely difficult to perform as it requires simultaneous intracellular recordings from Int1 and AB at elevated temperatures. Even many attempts at such experiments would likely result in a very low “*N*”.

– In the same vain, the sensitivity to temperature changes of the gastric rhythm has been studied here but with the pyloric network, being itself intrinsically sensitive to temperature changes, still active (Figure 3 and related text). What do we know on the intrinsic temperature sensitivity of the gastric rhythm when elicited by dpons stimulation but isolated from the pyloric network (AB neuron killed for example)?

Answered above. (Figure 4—figure supplement 2) Results.

– Data presented here show that coordination between PD and LG neurons is preserved after temperature increase, but that this is not the case between PD and DG neuron that shows no phase-coupling at high temperature (Figure 6). The PD neurons is used here as an indicator of the pyloric rhythm while the LG neurons indicates the gastric rhythm. Then what would be the conclusions of the authors if the DG neuron would have been used as the gastric rhythm indicator? How do you conciliate everything together?

The LG neuron is used as the gastric mill rhythm indicator because it is the motor neuron associated with the half-center oscillator that drives the gastric mill. Additionally, because the DG neuron is responsible for retracting the medial tooth, its timing is dictated indirectly by projection neuron input and LG feedback to MCN1.

Reviewer #3:[…] The main finding of the paper is that a previously-described integer-coupling between two rhythms remains more or less intact across temperature variations. It is a nice descriptive finding, but rather disappointing in that there is so much more that could have been done rather easily that would have given much more depth to this finding. Most obviously, because it is known that the source of the coupling is the inhibitory synapse from the pyloric pacemaker to the gastric mill half-center, it is quite important to know how the strength of this synapse affects the interaction at different temperatures. That is, to expand what Bartos et al., 1999, did across a range of temperatures. Short of that, it would have been nice at least to perturb the cycle period of the pyloric rhythm and see whether the interaction would remain robust across temperature despite changes in cycle period.

In order to study changes in synapse strength from AB to Int1, we would need stable recordings from both neurons across temperature, and as stated above this is very difficult if not impossible to do frequently enough to gather a complete dataset. Even in the Bartos et al., 1999 paper, only 4 preparations are recorded from at 11°C, likely reflecting the technical difficulties of these experiments.

While the study convinces the reader that integer coupling between pyloric and evoked gastric rhythms is robust to temperature changes, it does not attempt to explore the origin of this robustness, e.g. by using different methods to activate the gastric rhythm or even testing if integer coupling is present with spontaneous gastric rhythms (which they do analyze for another purpose).

The additional experiments added to this study (Figure 4—figure supplement 2) address this comment to some extent by studying how the gastric mill circuit is independently temperature robust in the absence of pyloric circuit input as a result of projection neuron input. As stated in the text, spontaneous gastric mill rhythms were not stable enough to determine whether or not they were integer coupled. This is why evoked rhythms were used in the first place. Results.

Because different methods of activating the gastric mill are likely to have different temperature sensitivities (e.g. comparing our results with those of Städele, Heigele and Stein, 2015), using different methods of driving gastric mill rhythms across temperature would not necessarily elucidate how or why the evoked rhythms we generate are temperature robust.